
# Bare-earth DEM Generation from ArcticDEM, and Its Use in Flood Simulation

**Yinxue Liu[1*], Paul D Bates[1], Jeffery C Neal[1]**

[1†] School of Geographical Sciences, University of Bristol, Bristol, UK

*Correspondence to*: Yinxue Liu (Yinxue.liu@bristol.ac.uk)

## Abstract

In urban areas, topography data without above ground objects are typically preferred in wide-area flood simulation, but are not yet available for many locations globally. High-resolution satellite photogrammetry DEMs, like ArcticDEM, are now emerging and could prove extremely useful for global urban flood modelling, however approaches to generate bare-earth DEMs from them have not yet been fully investigated. In this paper, we test the use of two morphological filters (Simple Morphological Filter-SMRF and Progressive Morphological Filter-PMF) to remove surface artefacts from ArcticDEM using the city of Helsinki (192 km$^2$) as a case study. The optimal filter is selected and used to generate a bare-earth version of ArcticDEM. Using a LIDAR DTM as a benchmark, the elevation error and flooding simulation performance for a pluvial event were then evaluated at 2 m and 10 m spatial resolution, respectively. The SMRF was found to be more effective at removing artefacts than PMF over a broad parameter range. For the optimal ArcticDEM-SMRF the elevation RMSE was reduced by up to 70% over the uncorrected DEM, achieving a final value of 1.02 m. The simulated water depth error was reduced to 0.3 m, which is comparable to typical model errors using LIDAR DTM data. This paper indicates that the SMRF can be directly applied to generate a bare-earth version of ArcticDEM in urban environments, although caution should be exercised for areas with densely packed buildings or vegetation. The results imply that where LIDAR DTMs do not exist, widely available high-resolution satellite photogrammetry DEMs could be used instead.

## 1 Introduction

The availability of an accurate bare-earth Digital Elevation Model (DEM) is important to many research fields, including identifying drainage related features and modelling flood inundation (Garbrecht and Martz, 2000; Yamazaki et al., 2014), deriving topography indices such as slope, orientation, and rugosity (Moudrý et al., 2018), estimating forest biomass and carbon (Jensen et al., 2016), and constructing 3D building heights (Marconcini et al., 2014).



For wide-area flood simulation in urban areas, a bare-earth DEM (i.e., a terrain model without
surface artefacts) is preferable in most circumstances to a Digital Surface Model (DSM) which
includes them. This is because the decision to include above terrain artefacts or not is a
consequence of the selected simulation resolution. Only when the simulation is conducted at
grid sizes allowing the resolution of building shapes and the street layout (typically < 5 m in
most urban topologies worldwide) does a DSM become useful. When aggregated to coarser
resolutions, the height of the surface artefacts contained in the DSM can block or alter flow
pathways in ways that lead to anomalous results when these data are used in hydrodynamic
modelling (Neal et al., 2009). Inundation simulations over regional and national scales usually
only become feasible with non-building resolving grid resolutions because of the exponentially
increased computational cost of running fine grid models and the limited availability of national
DEMs with resolutions finer than 5 m. Even at city and sub-city scales, non-building resolving
models may be preferable for ensemble and event set simulations (Mason et al., 2007; Schubert
and Sanders, 2012). As a result, bare-earth DEMs (also known as Digital Terrain Models or
DTMs) are essential for flood inundation simulations in urban areas and can also be beneficial
to a broad range of other research fields.
Unlike traditional, ground-based field survey, modern wide-area DEM collection
techniques rely on remote sensing from ground vehicle, airborne and satellite platforms. All
DEMs derived in this way include the heights of built-up area artefacts and vegetation to some
extent and require significant post-processing to obtain a bare-earth DEM. Commonly used
DEMs are collected using techniques including Interferometric Synthetic Aperture Radar (i.e.,
InSAR), optical stereo mapping and LIDAR. These different techniques, combined with the
platforms and the specific instrument characteristics, offer DEMs with varied coverage,
resolution, and accuracy (Lakshmi and Yarrakula, 2018; Zaidi et al., 2018). For example,
spaceborne and globally available InSAR DEMs offer wide coverage but they are constrained
by the geometry of the interferometric baseline and the temporal sampling of the spaceborne
platform and InSAR technique. The derived DEMs therefore have limited horizontal resolution
and accuracy (SRTM at ~30 m spatial resolution has reported mean absolute vertical error of
6 m, TanDEM-X at ~12 m spatial resolution has 90% linear error (i.e., LE90) in the vertical of
around 2 m) (Rodriguez et al., 2006; Wessel et al., 2018). Such vertical errors are significant
compared to the amplitude of most river flood waves, which typically range from 1-2 m up to
~12 m for the Amazon River at Manaus in Brazil (Trigg et al., 2009; Bates et al., 2013). Whilst
global InSAR DEM errors can be reduced by intelligent processing (O'Loughlin et al., 2016;



Yamazaki et al, 2017; Archer et al., 2018; Liu et al., 2021; Hawker et al., 2022) and by
aggregating to coarser grid resolutions to mitigate random errors, they remain distinctly sub-
optimal for much flood inundation modelling (Schumann and Bates, 2018). Instead, inundation
modelling is best conducted with DEMs generated using airborne LIDARs for most
applications. These have high accuracy, with a typical vertical RMSE of 0.05–0.2 m (Faherty
et al., 2020), and spatial resolution of 1-2 m such that they can identify the detailed structure
of floodplain geomorphology, buildings, vegetation, and important linear features such as flood
defences and their crest elevations. However, due to their (relatively) high cost of collection,
freely available LIDAR data only cover ~0.005% of the global land surface (Hawker et al.,
2018). DEMs derived from high-resolution stereo images, such as WorldView, have the
potential to cover the land surface globally with spatial resolution (and also perhaps accuracy)
comparable to LIDAR (Noh and Howat, 2015; Hu et al., 2016; Shean et al., 2016; DeWitt et
al., 2017). Whilst stereo photogrammetry was previously used to develop the (now superseded)
publicly available ASTER DEM (Hirano et al., 2003), more recent DEMs derived from high-
resolution photogrammetry such as WorldView, GeoEye, IKONOS and Pleiades images have
been kept as commercial products with a cost that is prohibitive for most academic studies.
However, the recent public release of an unprecedent resolution (2 m) satellite photogrammetry
DEM, ArcticDEM (Porter et al., 2018, https://www.pgc.umn.edu/data/arcticdem/), has brought
opportunities to explore the potential of such a product in flood inundation modelling.
ArcticDEM covers areas above 60°N and was produced using the SETSM method from in-
track and cross-track high-resolution (~0.5 m) imagery acquired by the WorldView and
GeoEye satellites. Using similar stereo-photogrammetry techniques, Google is also developing
a very high-resolution DEM using multiple satellite sources (Ben-Haim et al., 2019). However,
both products are DSMs and therefore contain surface artefacts which need to be removed to
enable their use in a range of geophysics applications including wide-area flood inundation
modelling. Previous research efforts to generate bare-earth terrain data from previously
released global DEMs such as SRTM and TanDEM-X have relied heavily on auxiliary data to
remove artefacts. For these next generation of high-resolution photogrammetry DEMs,
auxiliary data at comparable resolution to the DEM does not yet exist and different approaches
must be proposed.

Considering the high resolution of these photogrammetry DEMs, the algorithms

already developed to create bare-earth DEMs from LIDAR are likely to be applicable to this
task. For example, DeWitt et al. (2017) have shown that applying LIDAR filtering procedures


to a WorldView-generated DEM in densely vegetated areas can remove vegetation artefacts
and achieve a bare-earth terrain representation with accuracy comparable to LIDAR. Numerous
research studies have been conducted in the past decade to generate bare-earth DEMs (i.e.,
DTMs) from LIDAR point clouds (Sithole and Vosselman, 2004; Chen et al., 2007; Meng et
al., 2009; Zhang et al., 2016). Filtering strategies were reviewed by Chen et al. (2017), and
morphology-based filters were reported as robust and capable of removing non-ground objects.
Notably, Zhang et al (2003) proposed a progressive morphological filter (PMF) for removing
non-ground measurements from airborne LIDAR. The PMF method has subsequently
advanced by enabling automatic extraction of ground points from LIDAR measurements with
minimal human interaction and is now widely used as a base filter to classify ground and non-
ground points (Cui et al., 2013; Hui et al., 2016; Tan et al., 2018). Evolved from the
morphological filter idea, Pingel et al (2013) developed the Simple Morphological Filter
(SMRF) by designating the window size increasement strategy of the filter and employing a
computationally inexpensive technique to interpolate the non-ground pixels. The SMRF was
reportedly able to achieve low misclassification errors (2.97%) among 11 filter algorithms for
LIDAR DEM samples with various configuration of slope and artefacts and to be robust to the
algorithm parameterization (Zhang et al., 2016). However, despite previous research applying
LIDAR filtering strategies to WorldView photogrammetric DEMs (Rokhmana and Sastra,
2020), none of these filters has been tested on ArcticDEM and research about the performance
of different filters for removing surface artefacts from high-resolution photogrammetric DSMs
is also lacking, especially in urban areas.

Given their unprecedented resolution and potential wide-area coverage, bare-earth

photogrammetric DEMs can possibly be used to advance flood inundation simulation at
regional scales and beyond. Although at this stage the access to these DEMs is restricted, they
are very promising and could become an alternative to LIDAR data in the future as a result of
their much lower cost. This could especially benefit developing countries where wide coverage
of LIDAR data is likely to prove unaffordable for the foreseeable future. This research therefore
aims to develop an approach to generate bare-earth DEMs from ArcticDEM and to examine
the use of the data in flood inundation simulation. The proposed approach is expected to be
generally applicable to other high-resolution (~m scale) photogrammetry DEMs as well as
ArcticDEM. We first compare the ability of progressive and simple morphological filters (PMF
and SMRF) to generate a bare-earth DEM from ArcticDEM in the city of Helsinki, Finland by
evaluating the filtered ArcticDEMs against a reference bare-earth LIDAR data set. Next, for





the best performing filter a set of parameter combinations was applied to generate a realization
ensemble of filtered ArcticDEM, whose error metrics were then analyzed against the parameter
settings. We then use both the original ArcticDEM and filtered ArcticDEM realizations to
simulate a pluvial flooding scenario for Helsinki and compare these results to an identical
simulation using the LIDAR DTM. Pluvial flood simulation is a difficult for hydrodynamic
models even with excellent terrain data and therefore poses a rigorous and diagnostic test.
Lastly, limitations of the current research and future work that could further facilitate the use
of a bare-earth version of ArcticDEM in flood inundation simulation is discussed.

## 139 2    Data source and study site

ArcticDEM is stereo-photogrammetry DSM generated from in-track and cross-track
high-resolution (~0.5 m) imagery acquired by the DigitalGlobe constellation of optical imaging
satellites. The majority of ArcticDEM data was generated from the panchromatic bands of the
WorldView-1, WorldView-2, and WorldView-3 satellites. A small percentage of data was also
sourced from the GeoEye-1 satellite sensor. ArcticDEM is available in two formats: strip and
mosaic. Strip data is the output extracted by the TIN based Search-space Minimization
algorithm (Noh and Howat, 2015) and preserves the original source material temporal
resolution. Mosaic data is compiled from multiple strips that have been co-registered, blended,
and feathered to reduce edge-matching artifacts. Due to the errors in the sensor model, the
geolocation of the generated ArcticDEM has systematic offsets in the vertical and horizontal
directions which are reported in the product's meta-data.  Offsets for the mosaic data are
unknown so therefore the strip data set with the original horizontal resolution at 2 m (version
3.0) was used as the baseline DEM in this paper. The offset values of each strip data were
applied before generating the bare-earth ArcticDEM.
The city of Helsinki was selected as a study site for the following reasons: 1) both
ArcticDEM and a high accuracy LIDAR DTM are available at this site, with the vertical error
of the LIDAR DTM reported as 0.3 m; and 2) it is a typical urban environment with sparse to
medium density buildings mixed with large patches of vegetation; 3) as the most populated city
above 60ºN, the Helsinki metropolitan areas is very vulnerable to flooding. To standardize the
vertical reference system, the quasigeoid height was subtracted from ArcticDEM, converting
its reference system from WGS84 ellipsoid height to the Finland National Vertical Reference-
N2000 that is used for the LIDAR data. This conversion has an accuracy of 0.02 m.



Within the city of Helsinki two building-dominated samples (S1 and S2, both covering
areas of ~0.7 km$^2$) were chosen to compare the effectiveness of two selected morphological
filters: the PMF and the SMRF. Sample 1 is characterized by buildings with floor areas up to
10000 m$^2$, whereas smaller buildings (floor areas of ~500 m$^2$) are distributed throughout
Sample 2. A larger third sample (S3, which includes both S1 and S2) was selected to conduct
the bare-earth DEM generation and to assess the filter's performance in a complex urban
environment. Flood inundation modelling of the resulting DEM data was also performed over
sample area S3 (Fig. 1). The ArcticDEM strips data derived from WorldView-1 images
acquired on the 14$^{th}$ of March 2013 (WV01_20130314) and on the 16$^{th}$ of February 2015
(WV01_20150216) were found to cover most areas of S3 (92% and 99%, respectively).
Considering the possible bias caused by forest and snow, the ArcticDEM strips with source
images acquired during leaf-off seasons and under snow-free conditions are preferable. The
Finish forests are reported to be mostly evergreen with ~10% of deciduous trees (Majasalmi
and Rautiainen, 2021). The source images of both strips were acquired during leaf-off
conditions. The snow situation on the image acquisition dates was analyzed using the MODIS
NDSI_Snow Cover data (Hall et al., 2016). The acquisition date of the strip WV01_20130314
was found to be much less covered by snow compared to that of the WV01_20150216 strip.
Therefore, the strip WV01_20130314 was used as the main data source and areas within S3
which this strip does not cover or where voids were present were filled with data from the strip
WV01_20150216. These mosaiced strip data are shown in Fig. 1, with the extent of the two
strips displayed. The ArcticDEM for all samples in this paper refers to this mosaiced dataset.
Land use and land cover (LULC) for Helsinki was acquired from the CORINE Urban Atlas
2012 database (https://land.copernicus.eu/local/urban-atlas/urban-atlas-2012). This LULC
features 22 land cover types in Helsinki. In this paper, features were merged to four categories:
urban, forest, open land, and water. Details of this reclassification of the LULC data can be
found in Supplement Table S1.


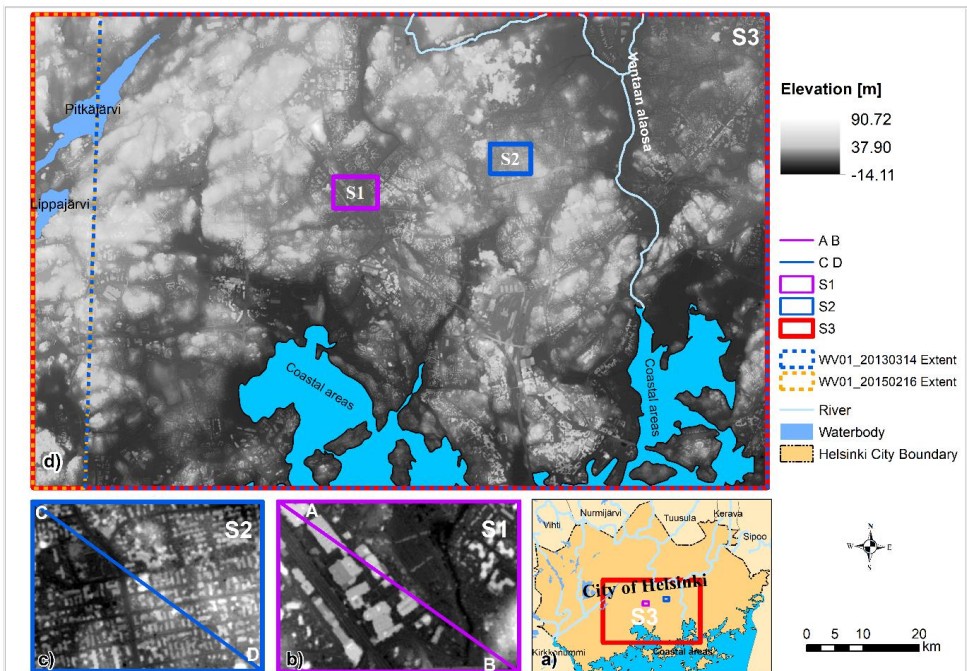


**Figure 1.** Locations of the three studied samples (S1, S2 and S3) within the city of Helsinki are shown at a).
Elevation values of the ArcticDEM at S1, S2 (overlain with transects crossing), and at S3 are shown in b), c), d)
respectively. Locations of coastal areas, lakes and rivers are also labelled. The ArcticDEM strip data is acquired
from the Polar Geospatial Center at https://data.pgc.umn.edu/elev/dem/setsm/ArcticDEM/mosaic/v3.0/2m/. The
water body outlines were acquired from the Finnish Environment Institute at
https://www.syke.fi/enUS/Open_information/Spatial_datasets/Downloadable_spatial_dataset.

## 3    Methods

### 3.1 Morphological filters

The generation of bare-earth ArcticDEM (our version of ArcticDEM with artefacts
removed) was conducted by employing two different morphological filters: PMF and SMRF
separately. They are considered because of their reported effectiveness in filtering LIDAR
point clouds, simple conceptualized parameters, and the fact that they are open access.

The PMF was designed to remove non-ground measurements (buildings, vegetation,
vehicles) from airborne LIDAR data (Zhang et al., 2003). It consists of an object detection and
an interpolation process which employs non-object pixel elevations to generate the values of
the object pixels. The PMF provides an advance on the morphological filter algorithm (Kilian





et al., 1996) by enabling a gradually increasing window width to detect non-ground objects
regardless of their size. In addition, an elevation difference threshold based on elevation
variations of the terrain, buildings, and trees was introduced to preserve the terrain. The
maximum window size and elevation variation threshold parameters control the filtering
process (more details can be found at Zhang et al., 2003).

More recently, a SMRF was proposed by Pingel et al (2013), also with the aim of
removing non-ground measurements from airborne LIDAR data. While the SMRF follows a
similar two-step process to the PMF, the approaches taken to detect objects and interpolate
elevation values of objects are different. SMRF adopts a linearly increasing window (as
opposed to the exponential increase of PMF) and simple slope thresholding, along with a novel
image inpainting technique. Like the PMF, the maximum window size ($W_{max}$) and slope
threshold ($S$) (equivalent to the elevation variation threshold of PMF) parameters control the
performance of the filter (Fig. 2). The core of the filter is the object detection where
morphological opening is applied to the original surface based on the current window size ($W_i$)
increasing from one pixel, by one pixel, to the maximum window size (in distance units, meters
in this research). For each window size within the range, the difference between the original
surface ($W_i=1$) or the surface from the last step ($W_i>1$) and the morphologically opened surface
is calculated and this difference (for example, $d_0$, $d_1$, $d_2$ in Figure 2) is compared with the
current difference threshold ($D_i$) (defined as the slope threshold $S$ multiplied by the current
window size $W_i$) to determine whether the object flag of the pixel should be accepted or
rejected. When the difference is smaller than the current difference threshold ($D_i$), the object
flag of these pixels is rejected (Fig.2 III) and the elevated areas are retained. Otherwise, pixels
are flagged as objects and then interpolated (Fig.2 I, II). When the maximum window size is
smaller than the patch size of the elevated areas (for example, $l_3$), the morphological opening
will be unsuccessful, and elevations in that patch area remain almost identical to the original
elevation (Fig.2 IV).

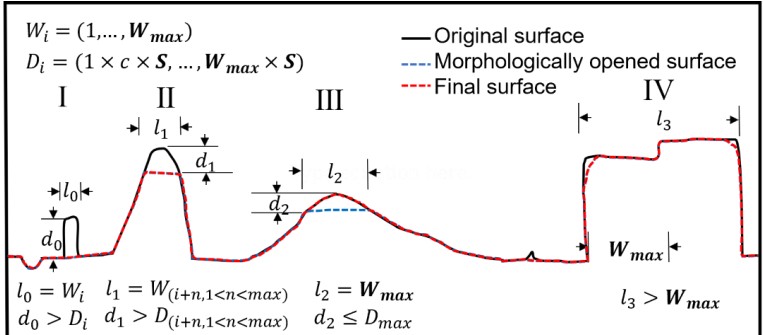

**Figure 2.** Illustration of the SMRF filtering process in a simplified urban environment with artefacts (I, IV) and hills (II, III). The symbols are W: window size, D: difference threshold, C: cell size (C equals 2 m in this case), S: slope threshold, l: patch size of the elevated areas.

### 3.2 Optimal filter selection and error evaluation of the ArcticDEM-SMRF realizations

At Sample S1 and S2, combinations of a range of window size (i.e., maximum window size) and slope threshold parameters were tested for both the PMF and SMRF filters (Table 1). The optimal filter was identified as the resultant DEMs with the smallest error (Root Mean Square Error, i.e., RMSE) filtered using PMF and SMRF respectively (details are presented in Sect. 4.1). Then, the best performing filter (SMRF) was applied to Sample S3 with a range of window size and slope threshold parameters (Table 1), which generated a total of 234 filtered ArcticDEM realizations, hereafter called ArcticDEM-SMRF. Using the LIDAR DTM as the reference, the RMSE and Mean error of the ArcticDEM-SMRF realizations as well as the reduction of RMSE over the original ArcticDEM-SMRF was calculated at pixel level (2 m) (Eq. (1)-(3) and Text S1 in the Supplement ). Due to other possible error sources, like shadow effects in the photogrammetry DEM, the calculations excluded values outside the 2.5th and 97.5th percentile as outliers. The ArcticDEM-SMRF with the lowest RMSE for all land areas among the realizations is termed the optimal ArcticDEM-SMRF. The three error metrics of the ArcticDEM-SMRF realizations were analyzed against the window size and the slope threshold parameter to examine the effectiveness of the SMRF filter at removing artefacts. As the artefacts of S3 are a mixture of buildings and vegetation, the filter effectiveness to these parameters was analyzed separately for all land areas, only urban areas, and only forest areas.





***Table 1.*** *Key parameter settings of the morphological filters tested in the three samples.*

| Filter | Sample | Window size (m) | | Slope threshold | |
|---|---|---|---|---|---|
| | | range | interval | range | interval |
| PMF | S1 | 10-66 | 4 | 0.1-0.3 | 0.2 |
| | S2 | 10-66 | 4 | 0.1-0.3 | - |
| SMRF | S1 | 10-50 | 2 | 0.01-0.1 | 0.005 |
| | S2 | 10-50 | 2 | 0.01-0.1 | 0.005 |
| | S3 | 10-180 | 10 | 0.03-0.15 | 0.01 |

* The unit of the slope threshold values shown here is radian for PMF, percent for SMRF.
3.3 Flood inundation evaluation of the ArcticDEM-SMRF realizations
For the 192 km$^2$ area covered by Sample 3 simple pluvial models were built at 10 m
spatial resolution instead of the original 2 m of the ArcticDEM due to computational cost
considerations. These models use DEM inputs from the LIDAR DTM, the original ArcticDEM,
and the ArcticDEM-SMRF realizations which were filtered with various parameter
combinations of the SMRF filter, respectively. The LIDAR DTM simulation was used as the
benchmark. For this computation the hydrodynamic model LISFLOOD-FP was used (Bates et
al., 2010). The model solves the local inertial form of the shallow water equations in two
dimensions across the model domain. For pluvial flood modelling, the model takes the terrain
elevation and rainfall data as inputs, and uses a raster-on-grid approach to calculates the
velocity, water depth, and inundation (Bates et al., 2021). The input DEMs were aggregated to
10 m by averaging before being used in the flood simulation. For the ArcticDEM and
ArcticDEM-SMRF models, elevation values in coastal areas (covered by water) were replaced
with the LIDAR DTM values. This was done to remove the impact of the DEM error in non-
land areas on the simulation. Rainfall data were acquired from the Climate Guide of Finland at
https://www.klimatguiden.fi/articles/database-of-design-storms-in-finland. It provides the
database of design storms with the real momentary variations in intensity for locations across
Finland. This database was generated based on radar measurements and derivations. An
extreme rainfall with a duration of 3 h and a return period of 500 years was used in the
simulation. This was selected to minimize the simulation time while ensuring that the
difference between the simulations was distinguishable. Under this duration and return period
conditions, the precipitation data at the nearest station (60.04ºN, 102.54ºE) to the city of
Helsinki was used. The precipitation is 102.54 mm in total with peak intensity at 182.4 mm/h.


The simulation results were compared to the LIDAR DTM benchmark in terms of the
simulated flood extent using the Critical Success Index (CSI) score, the Hit Rate, and the False
Alarm Ratio (FAR) defined by Eq. (1) - (3) (Wing at al., 2017), and the water depth errors
using the RMSE and the Mean error, Eq. (4) and (5). A wet cell is defined as one with simulated
water depth exceeding 0.1 m in this paper. As is typical in often the case in pluvial simulations,
small isolated wet areas (where the number of connected wet cells was less than 15) were
excluded from both the benchmark model (LIDAR) and the evaluation target models
(ArcticDEM and ArcticDEM-SMRF) before calculating the metrics. First, all five metrics
using the set of ArcticDEM-SMRF DEMs derived using different filter parameters were
compared with the flooding performance of the original ArcticDEM. Then, the relationship
between the five flooding metrics and the RMSE and Mean error of the DEM of the
ArcticDEM-SMRF realizations (aggregated at 10 m) was depicted for all land areas, urban and
forest areas individually. Furthermore, the flooding performance simulated by the optimal
ArcticDEM-SMRF was evaluated spatially.
$CSI = \frac{A}{A+B+C}$ (1)
$Hit\ Rate = \ 100\% \times \frac{A}{A+C}$ (2)
$FAR = \ 100\% \times \frac{B}{A+B}$ (3)
$RMSE_{water\ depth} = \sqrt{\frac{\sum_{i=1}^{i=n}(WD_{i,c,DEM}-WD_{i,c,LIDAR})^2}{n}}$ (4)
$Mean\ error_{water\ depth} = \frac{\sum_{i=1}^{i=n}(WD_{i,c,DEM}-WD_{i,c,LIDAR})}{n}$ (5)
*A is the number of pixels which are wet in both the DEM and the LIDAR simulation, i.e., where the two models
agree; B is the number of pixels which are wet in the DEM simulation but not the LIDAR simulation, i.e.,
overestimation; C is the number of pixels which are wet in the LIDAR simulation but not the DEM simulation,
i.e., underestimation.
*$WD_{i,DEM}$ is the water depth at pixel i simulated using the DEM (ArcticDEM-SMRFs or the original ArcticDEM
depending on the calculation target), and n is the number of the wet cells (wet in either the LIDAR or the DEM
simulation) within category C. Category C is defined by the land use and land cover, and they can be all land
areas, urban, forest. For example, the water depth RMSE of ArcticDEM-SMRF in urban areas are calculated based
on the ArcticDEM-SMRF pixels within urban areas.



**4      Results**

4.1 Optimal filter selection

The effect of using the PMF and SMRF filters to remove artefacts from the ArcticDEM

in the two building-dominated samples S1 and S2 is evaluated by plotting the error distribution
and transect profiles. The filtered ArcticDEM with the smallest RMSE using each filter's
optimum parameters is shown in Fig. 3. The optimal PMF parameters for S1 and S2 are window
size = 42 m, 30 m, slope threshold = 0.3 (radian) for both, and the optimal SMRF parameters
for S1 and S2 are window size = 32 m, 14 m, slope threshold = 0.08, 0.05 (%), respectively.
The calculation of error figures was conducted at 2 m pixel scale.
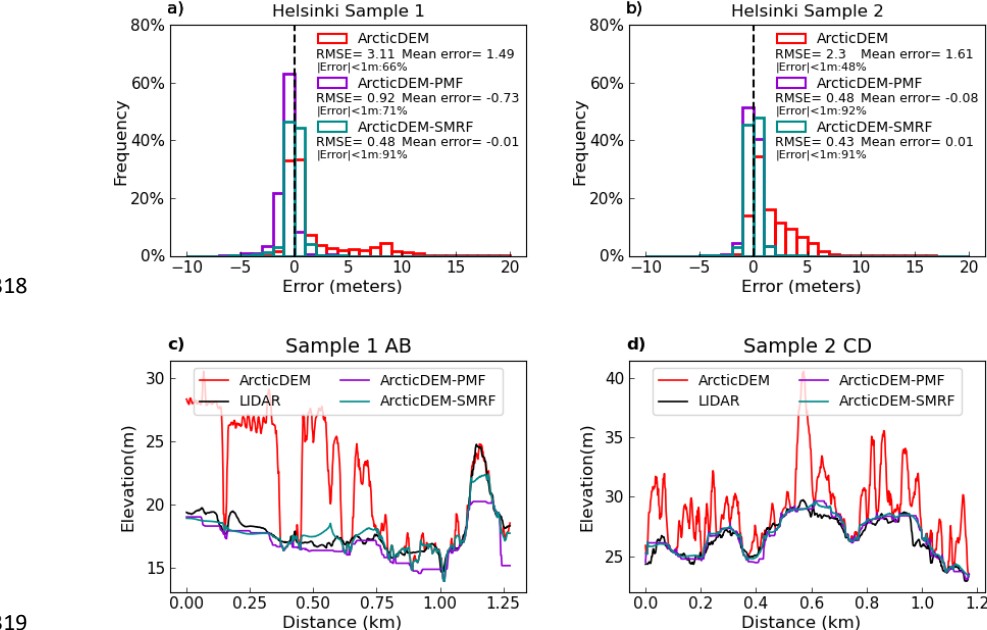


**Figure 3**. Error histograms of ArcticDEM, ArcticDEM with PMF applied (ArcticDEM-PMF) and ArcticDEM
with SMRF applied (ArcticDEM-SMRF) for sample S1, a) and S2, b). Profile of ArcticDEM, ArcticDEM-PMF,
ArcticDEM-SMRF, and LIDAR DTM for transects through S1, c) and S2, d). The location of transects is shown
in Fig. 1b and c.

The error histograms show that both PMF and SMRF can effectively remove much of

the bias caused by artefacts in ArcticDEM, with the resulting RMSE falling below 1 m in all
cases. The count of pixels with error <1 m increased to 91% in both samples. The SMRF filter
achieved a lower RMSE (0.48 m and 0.43 m for S1 and S2, respectively) compared to PMF





(0.92 m and 0.48 m) (Fig. 3a and b). The Mean error of the filtered DEMs for S1 and S2 also
evidences that SMRF has an advantage over PMF.

The DEM profile through S2 shows that SMRF and PMF work similarly well, while

the profile through S1 shows that SMRF can preserve more terrain details than PMF in
moderate hillslope areas (Fig. 3c, e.g., distance 0.75-1.0 km). However, both filters incorrectly
identified the steepest areas of S1 as artefacts, especially PMF (Fig. 3c distance 1.0-1.25 km).
Considering both the histogram and profile results, SMRF was selected as the optimal filter to
remove the artefacts from ArcticDEM for this site.

The sensitivity of the slope threshold and the window size parameter to the error metrics

for ArcticDEM-SMRF at sample S1 and S2 can be found in the Supplement Figure S1 and
Text S2.

4.2 Bare-earth DEM generation and its error evaluation

In order to understand the effectiveness of the SMRF in a more complex urban

environment the error metrics RMSE, RMSE reduction percentage and Mean error of the
ArcticDEM-SMRF realizations were computed for the larger sample S3. These metrics were
analyzed against the window size and slope threshold parameter of the SMRF filter to evaluate
the sensitivity of ArcticDEM-SMRF error to changes in these values. As the surface artefact
bias in S3 is mainly caused by buildings and forests, the analysis was conducted for all land
areas as well as for urban areas and forest areas separately (Fig. 4).

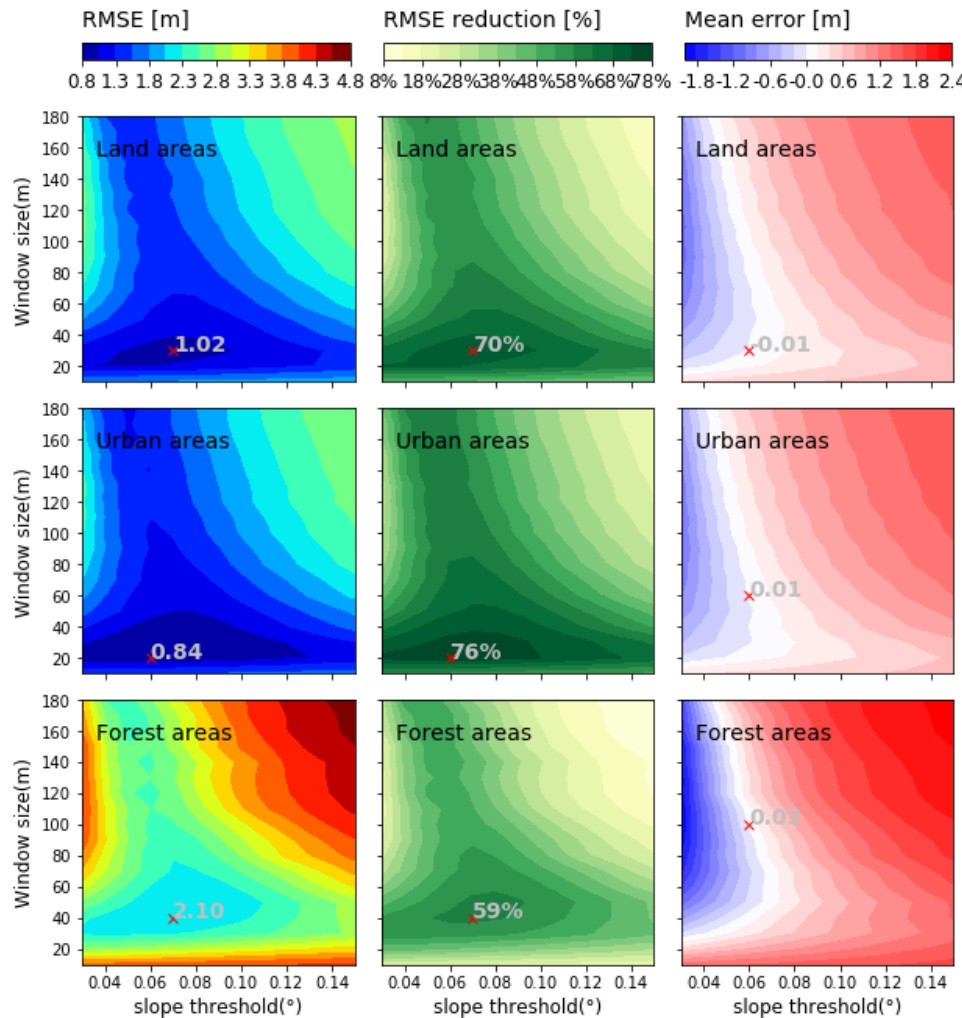


**Figure 4**. Surface plots of the slope threshold and the window size parameters of the SMRF filter against the

RMSE, the RMSE reduction percentage and Mean error of the filtered DEM-ArcticDEM-SMRF for sample S3.

The location of the smallest values of the RMSE, the greatest values of the RMSE reduction and the smallest

absolute values of the Mean error are marked as red crosses, with the values displayed. Parameter details can be

found in Table 1.

For area S3, the smallest RMSE of the ArcticDEM-SMRF realization is 1.02 m (i.e.,

the optimal ArcticDEM-SMRF) within all land areas, 0.84 m in urban areas and 2.1 m in forest

areas. These values represent 70%, 76% and 59% reductions of the ArcticDEM error

respectively. The greatest reduction was achieved with a slope threshold of 0.07 combined with

a window size of 30 m for all land areas or 40 m for forest areas, and a slope threshold of 0.06





with a window size of 20 m for urban areas. Although the RMSE of the optimal ArcticDEM-
SMRF is greater than that computed for samples S1 and S2 (Fig. 3a, b), the magnitude of the
error reduction indicates that the SMRF is still very effective at removing surface artefacts from
ArcticDEM for this larger sample. More than 40% of the 234 parameter combinations can
reduce the RMSE by greater than a half. Thus, the SMRF filter is considered as a robust filter
given that the tested parameters range are set generally broad.
This robustness also means that different combinations of window size and slope
threshold can achieve similar resultant RMSE (for example, for urban areas window size = 20
m with slope threshold between 0.03 and 0.12, or window size = 40 m with slope threshold
between 0.05 and 0.1). For sample S3, the most effective window size ranges from 20 m to 30
m for all land areas, from 20 m to 40 m for urban areas, and from 30 m to 60 m for forest areas
with slope threshold between 0.04-0.1. From the parameter selection perspective within the
effective range, a smaller window size is more robust and is therefore preferred because the
choice of the corresponding slope threshold is broader compared with a larger window size.
When the window size is smaller than 20 m, the error of the filtered DEM becomes almost
independent from the slope threshold parameter choice. With some parameter combinations
the SMRF becomes less effective at removing artefacts or introduces negative errors, which is
a combination of large slope threshold (> 0.1) and large window size (> 60 m) or when the
slope threshold is smaller than 0.04 with window size larger than 20 m. Additionally, when the
window size parameter is above 60 m, the Mean error of the filtered DEM becomes more
sensitive to the slope threshold, especially with slope threshold smaller than 0.06.
The error distribution of the optimal ArcticDEM-SMRF was also analyzed spatially
and statistically (Fig. 5).
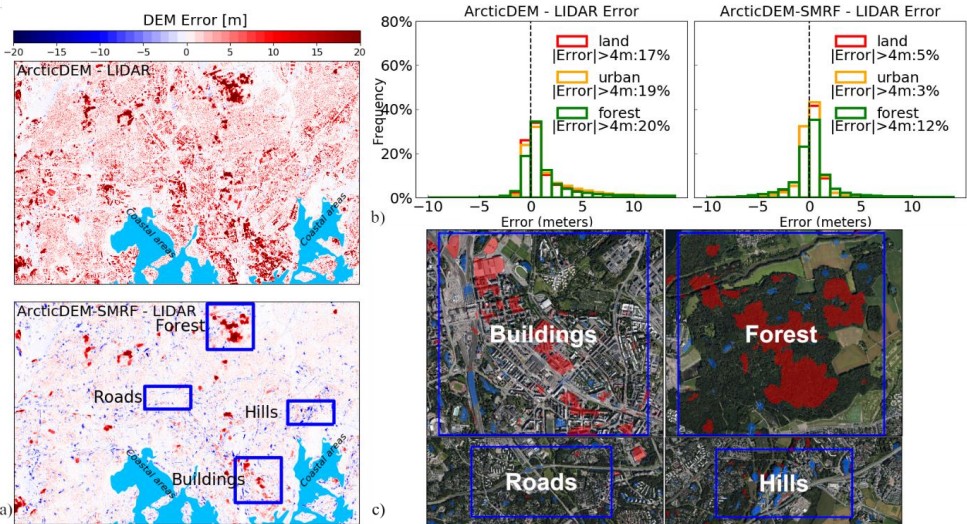


**Figure 5**. a) Difference maps between the original ArcticDEM, the optimal ArcticDEM-SMRF (with slope
threshold = 0.07, window size = 30 m as the SMRF parameters) and the LIDAR DTM at 2 m. b) The error
histograms of the original ArcticDEM, the optimal ArcticDEM-SMRF, where the calculation was conducted at
2 m pixel level. In the bottom map of a), example locations of four features that relate to the residual errors of
the ArcticDEM-SMRF are labelled. The aerial image of these locations is shown in c) where areas with errors
exceeding 4 m were marked (> +4 m as red polygons and < -4 m as blue polygons, in 50% transparency). The
aerial image is orthophotograph of Helsinki with a horizontal resolution at 8 cm, acquired during growing
season of 2017, which was accessed from Helsinki Region Infoshare at
https://hri.fi/data/en_GB/dataset/helsingin-ortoilmakuvat.

The error maps before and after applying the filter show that the SMRF method largely
reduces the errors in ArcticDEM, especially in urban areas (Fig. 5a, b). Although some residual
errors (> 4 m) are present in the optimal ArcticDEM-SMRF, they comprise a very small
percentage (~5%) of the whole area (Fig. 5b). Errors in dense forest areas and for closely spaced
buildings with large floor areas typically present as the largest positive residual errors as shown
in Fig. 5c. Large negative errors occur in hillslope areas (usually slope >10°) and in some areas
where above-ground traffic links such as junctions, viaducts, or overpasses are present (Fig.
5c).

### 4.3 Flood inundation evaluation of the ArcticDEM-SMRF realizations

The flooding evaluation metrics simulated using the original ArcticDEM and the
ArcticDEM-SMRF realizations for all the 234 parameter combinations are plotted against the
402 DEM error metrics (RMSE, Mean error calculated at 10 m) for each DEM realization in Fig.

6. This analysis was conducted for all land areas, urban and forest areas separately.

**Figure 6**. Surface plot of the CSI score, Hit Rate, FAR, the water depth RMSE and Mean error (ME) simulated

using the ArcticDEM-SMRF realizations (ArcticDEM filtered using the 234 SMRF parameter combinations) at

sample S3 plotted against the RMSE and the Mean error of each realization member. The location of the highest


CSI and Hit Rate, the smallest FAR, RMSE and the smallest absolute value of mean water depth error are

marked as red crosses, with the values displayed. In addition, the RMSE, Mean error of the original ArcticDEM

are located and marked as blue crosses in each panel with the five metrics value of the original ArcticDEM

simulation displayed.

As a result of the reduced RMSE and Mean error the flooding performance of

ArcticDEM-SMRF improved for almost all the parameter combinations. For the whole S3 area,
the CSI score increased by 0.19, achieving a maximum value of 0.56 against the benchmark
LIDAR simulation. CSI increased by 0.17 in urban areas (to 0.49), and by a slightly smaller
amount of 0.13 in forest areas (to 0.49). It should be noted that although residual errors of
ArcticDEM-SMRF in urban areas are not as large as in other areas, the flooding extent
prediction skill doesn't exceed a CSI of 0.5. This is likely because the flooding extent for a
pluvial simulation becomes very sensitive to the small-scale errors of the DEM in flat areas
where water depths are typically extremely shallow. In this sense, simulation of pluvial
flooding is a rigorous test of DEM quality and the results achieved here using ArcticDEM-
SMRF should be interpreted with this in mind. It is also important to remember that the LIDAR
data, whilst good, is not truth, and has a reported vertical error of 0.3 m.  LIDAR noise and
systematic error also contribute to some of the difference between the flooding performance of
models using the LIDAR and ArcticDEM-SMRF data.  Simulations of fluvial flooding, where
depths are typically greater, would likely score higher on the spatial extent performance
metrics. The Hit Rate was improved by an even larger amount: 24, 24 and 18 percentage points
in all land areas, urban areas, and forest areas, respectively. The FAR was reduced by 5
percentage points in all land and urban areas, 3 percentage points in forest areas. The greater
improvement in urban areas provides evidence that the filter is especially effective at improving
the flood simulation in urban areas, considering that flooding in urban areas is usually more
fragmented and thus is more difficult to predict than in forest areas. With the ArcticDEM-
SMRF, the simulated water depth error (RMSE) was reduced by up to 0.11 m for all land areas
and urban areas compared to the original ArcticDEM, and this reduction was slightly smaller
(0.06 m) in forest areas. Although the water depth is still underestimated, the ArcticDEM-
SMRF simulation reduced the average error by 0.12 - 0.17 m compared to that of the original
ArcticDEM. Unlike the flooding extent performance comparison between urban and forest
areas, the water depth error in urban areas is always smaller than in forest areas in both the
simulation with the original ArcticDEM and the ArcticDEM-SMRF realizations. This is a
result of the smaller DEM error in urban areas. Thus, it can be inferred that the water depth


error is more sensitively impacted by the error of the DEM than the flood extent, at least in the
case of these pluvial flooding simulations.

Unsurprisingly, the ArcticDEM-SMRF with the smallest vertical elevation error

achieved the best flooding performance for all land areas. However, there are two other cases
where equally good flooding performance can be simulated using ArcticDEM-SMRF with
larger error. The first case occurs when the DEM is over-corrected by the filter, i.e., where
negative errors are present in the filtered DEM. In this case, some steep areas are identified as
objects and are flattened incorrectly. As these are not prone to be flooded, the flooding
performance is barely impacted. The second case occurs when the DEM preserves the most
terrain details, shown at the spike areas in Fig. 6 (ArcticDEM-SMRF mean error of >-0.5 m
and CSI between 0.54 and 0.59 for land areas). This implies that for flood simulation the
filtering strategy can perform equally well by aiming to achieve the lowest DEM error, or by
removing the artefacts as much as possible (over-filtering), or by preserving the terrain details
as much as possible (filtering with a small window size of 10 m in this case study).

The spatial distribution of the flooding extent and water depth error simulated using the

optimal ArcticDEM-SMRF is shown in Fig. 7.

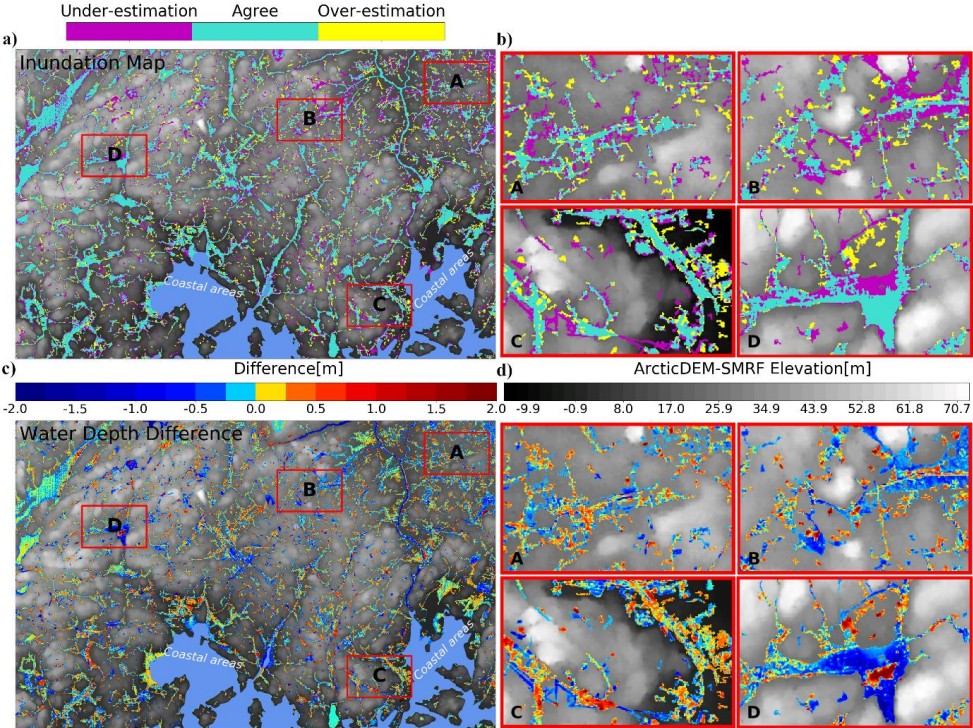






**Figure 7**. Inundation extent simulated using the optimal ArcticDEM-SMRF parameters (slope threshold = 0.07, window size = 30 m) at 10 m, where inundation areas that agree with, overpredict and underpredict the extent of the LIDAR DTM 10 m simulation are shown at a). The water depth difference between the ArcticDEM-SMRF and LIDAR DTM simulations for all wet cells is shown at c). Areas with significant disagreement are marked by rectangles denoted A, B, C, D with the zoomed in maps displayed at b) and d). The land cover of A and C is building-dominated, and forest-dominated at B and D.

For a 10 m spatial resolution simulation, ArcticDEM-SMRF can capture the major flooded areas correctly with underestimation mainly around the edge of the agreed wet cells and with overestimation presenting as scattered, small patches. Total underestimated area was about 1.8 times greater than that of overestimated areas. Underestimation disproportionately occurred along traffic links and along the edge of streams, in lake areas as well as in some of the forest areas with significant residual errors (Fig. 7a).

Unlike the general underestimation for the domain as a whole, both underestimation and overestimation were present in urban areas and the number of pixels that are under- and over-estimated is similar. These errors appear as disconnected patches with smaller size and their spatial distribution is more even compared to errors in forest areas (Fig. 7b-A, C in contrast to Fig. 7b-B, D).

The greatest water depth error is present in forest areas (Fig. 7d-B, D) where the ArcticDEM-SMRF simulation either fails to inundate these areas (underestimation) or generates much shallower water depths compared to that simulated using the LIDAR DTM. In urban areas, the water depth error simulated using the ArcticDEM-SMRF is relatively small, varying between -0.5 m and 0.5 m (Fig. 7d-A, C).

## 5 Discussion

### 5.1 The selection ArcticDEM strips

The error of different ArcticDEM strips covering the same areas could vary significantly. In this study site, we found that the main difference in error occurs in forest areas. Within a selected 11 km$^2$ forest area the error of the strip acquired on the 16th of February 2015 is 12.2 m, while within the same area that of the strip acquired on the 14th of March 2013 was much smaller (6.66 m). From air photos, no noticeable forest coverage change was found within the selected areas between the acquisition years of the two strips. Therefore, the difference between strips could be caused by the leaf-on/off differences or the snow situation. In this case, since both acquisition dates are during leaf-off season it is likely a result of



differences in snow cover. Even for the building dominated samples, the error at S1 and S2 of
the former strip (acquired on the 16th of February 2015) is 0.31 m, and 0.88 m larger than the
latter strip. Thus, we suggest that for general bare-earth generation from ArcticDEM, different
strips should consider the forest characteristics (evergreen or deciduous) and the weather
conditions (snow free or not) on the data acquisition date in overlapping areas. Strip data in
leaf-off and snow-free conditions will represent more of the ground elevation compared to data
collected in leaf-on or snow-covered conditions. Also, snow-free condition avoids the feature
matching difficulty between stereo images in the DEM generation process, which happens
often because the presence of snow results in low-contrast and repetitive image textures (Noh
and Howat, 2015). The snow condition on the strip data acquisition date can be checked using
the daily MODIS snow index product (Hall et al., 2016).
5.2 SMRF filter parameters
A direct application of the SMRF filter proved to be effective at removing most of the
surface artefacts at this study site, especially for buildings. It means that this LIDAR processing
tool can be employed without modification in generating a bare-earth ArcticDEM in urban
areas with buildings spacing at medium density like Helsinki (0.22 floor area ratio on average
within a 250 m grid cell, https://hri.fi/data/en_GB/dataset/rakennustietoruudukko). The SMRF
is generally robust to its window size and slope threshold parameter choices with respect to the
error reduction of the filtered ArcticDEM and the reduction could be optimized by narrowing
the parameters to certain values. Although by the algorithm definition, the parameters should
be set as the largest patch size and the greatest terrain variation, this research shows that in a
large domain application the window size and the slope threshold parameter range should be
gauged around the median value of the artefacts patch sizes and of the terrain variation values.
At this study site, the range of the window size is 20 - 40 m and a range of 0.04 - 0.1 for the
slope threshold performed best, with optimal values located at the median point of the
distribution. Within the reasonable range, a smaller window size proved to be more robust in
that it will be less sensitive to the choice of the slope threshold.
When benchmarking to a LIDAR DTM simulation, similarly good flood simulation
performance for the filtered DEMs is found to be achieved by the ArcticDEM-SMRF with
smallest error, or negatively biased ArcticDEM-SMRF or positively biased ArcticDEM-SMRF
preserving the most terrain details. Applying the SMRF filter is a trade-off between the removal
of artefact errors and the loss of terrain detail. When the SMRF is applied with a small window


size (such as 10 m), most of the terrain details can be maintained in the ArcticDEM-SMRF
while the residual error of the DEM can be large as a result of the residual artefacts with large
patch sizes. Since these preserved terrain details might be important in the inundation
simulation, the flood performance could be better in some places than when more of the
residual errors are removed at the cost of losing these details. Whilst the SMRF filter tends to
produce negative errors on hillslopes, these areas are not flooding-prone so the flooding
inundation is not significantly affected. The error sensitivity of the ArcticDEM-SMRF
realizations to the SMRF parameters at different slope areas is included in the Supplement as
Figure S2 and Text S3. Despite the above points, the filter parameters of the two latter cases
are not easy to gauge and likely to varying from location to location, thus using the median
values of the artefacts size and terrain variation is suggested.
5.3 Limitations
Although the SMRF filter successfully removed most of the ArcticDEM errors caused
by artefacts, there is a small percentage of artefact errors (~5%) that remains in dense built-up
areas and in large vegetation patches. Pixels in these areas are not entirely flagged as objects
with a window size of 30 m and some pixels are instead wrongly designated as 'ground' values
in the interpolation. Even though with an enlarged window size the remaining artefact errors
could be removed by the SMRF, the interpolation over large patch areas would potentially be
unsuccessful due to a lack of ground elevations within these zones. Additional data or a tailored
approach is required to achieve the desired result in areas with large patch sizes. For building
artefacts, the OpenStreetMap building footprint data could be helpful to predefine the areas of
objects. The ICESAT2 terrain elevation might be useful to provide additional ground elevations
in forest areas with large patch sizes (Neuenschwander et al., 2020; Tian and Shan, 2021).
With this filter, artefacts with small size are usually identified before the window size
reaches the maximum and the subsequent interpolation is also more successful. This makes the
SMRF filter more effective at removing building artefacts than vegetation due to the general
smaller size of building patches. However, some desired features that present similar elevated
characters to building artefacts (such as traffic junctions or levees) might be removed by the
filter unfavorably, and negative errors are shown in these areas. It becomes very tricky to
preserve these feature heights by any automatic filtering approaches without the location
information of the features. With more sophisticated method, likely with some ancillary data,
this could be possible (Wing et al., 2019). For hilly areas, some of the natural terrain might be



identified as artefacts by the SMRF incorrectly and the subsequent interpolation can cause the
loss of terrain details. The error histograms and analysis of the ArcticDEM-SMRF generated
with different window size parameters at buildings and forest with large patch size, hillslope,
and roads examples can be found in Figure S3 and Text S4 in the Supplement. Thus, in terms
of the bare-earth DEM generation, the filter is likely to be less effective for areas with densely
packed artefacts or hilly areas.
For flood simulation the errors in ArcticDEM-SMRF along river channels and over
floodplains is particularly critical, and further DEM processing here could lead to additional
improvements. In the ArcticDEM-SMRF, the elevations of the river sections that run through
large patches of forest are positively biased because of the reduced effectiveness of the SMRF
filter in these areas. The water depth error along the river network is expected to be mitigated
once these blockages are removed, such as by using quantile regression techniques
(Schwanghart et al., 2017). Similarly, elevation values along the road network (acquired from
OpenStreetMap) were particularly interesting and extracted for further analysis. It was found
that the SMRF filter largely lowered the elevation of the road network where artefacts are
present. But the resulting DEM from SMRF is interpolated based on all neighbouring pixels
and not only along the road pixels on either side of the artefact removed. Thus, an unsmooth
distribution of the along-road elevation was generated, which is not ideal for flood simulation
and likely to be inaccurate. A linear interpolation along the central line of the road network
with a buffering around that could be used to reduce these errors in the future. It should be
noted that the buffering width of the central line of roads could be tricky to define when there
is not accurate road width data available.
Moreover, sinks can be present in ArcticDEM (areas with substantially lower elevation
than neighbouring pixels), possibly because of the shadow effect which is a common issue for
photogrammetry DEMs (Noh and Howat, 2015). These sinks should be identified and filled in
future work.
**6      Conclusions**
In this paper, we examine two morphological filters (PMF, SMRF) for removing
surface artefacts from the ArcticDEM strip data in a complex urban environment using the city
of Helsinki as a case study. We then assess the improvement in flood inundation simulation
provided by the filtered ArcticDEM relative to a LIDAR DTM benchmark in a pluvial flooding
scenario. To our knowledge, it is the first examination of the approach to generate bare-earth


ArcticDEM data specifically for flood applications. It was found that the SMRF performs better
at removing surface artefacts from ArcticDEM than the PMF filter, and it is robust to its
parameter setting. The optimal parameter combination is around the median value of the patch
size distribution of the artefacts and of the terrain variation, which resulted in an optimal
window size of 30 m and slope threshold of 0.07 in the city of Helsinki. With SMRF, the overall
error of the ArcticDEM can be reduced by up to 70% with the optimized parameters, achieving
a final RMSE of 1.02 m.

The flood inundation simulation performance of a standard two-dimensional
hydrodynamic model was considerably improved when using the filtered ArcticDEM in that
40% of the underestimated areas simulated by the ArcticDEM were eliminated. Although the
flooding extent performance simulated by the ArcticDEM-SMRF is still not a strong match to
the LIDAR DTM benchmark (CSI=0.56, although some of this difference will be caused by
errors in LIDAR itself), the pluvial flood simulation should be seen as a rigorous test as the
inundated areas usually vary within few pixels in urban areas and are easily impacted by small-
scale errors. The simulated water depth error of the optimal ArcticDEM-SMRF model is
comparable to the likely error of the LIDAR DTM simulation, as a result of ~0.1 m
improvement comparing to the original ArcticDEM.

The residual errors of the filtered ArcticDEM are mainly composed of: 1) positive
errors for artefacts with large patches sizes, which are not entirely removed by the filter; and
2) negative errors in hilly areas which are incorrectly identified as artefacts. Thus, when using
the SMRF filter in other study areas where the artefacts have a much higher density or artefacts
with a large patch size comprise a significant proportion of the study area, the effectiveness of
the SMRF filter could be less significant compared to the results of this study. Some
modification of the SMRF filter might be able to remove the densely distributed artefacts and
auxiliary data are likely to be needed to guarantee satisfying interpolation results. Applying the
SMRF filter to hilly areas is also likely to yield a less effective performance. From the
perspective of flood inundation simulation, the SMRF parameters should be configured
towards optimizing their range to generate the DEM with the lowest error.

This paper suggests that applying the SMRF without any algorithm modification is
effective to generate bare-earth DEMs from ArcticDEM and are likely to be applicable to other
high-resolution photogrammetry DEMs and other application areas. The generated bare-earth
DEM shows largely reduced error comparing to the original ArcticDEM and comparable
simulated water depth error to the LIDAR benchmark. Thus, it is a promising alternative to
LIDAR data for locations where such data are either not available or would not be cost efficient.
In the future, using ancillary data to address the residual errors of the filtered DEM should be
integrated to the bare-earth ArcticDEM generation process. To facilitate the use of bare-earth
ArcticDEM in flood simulation, the blockage of residual error within rivers and errors along
road network should be carefully treated.

**Data and code availability**

LIDAR        data        at        2        m        was        acquired        from
https://tiedostopalvelu.maanmittauslaitos.fi/tp/kartta?lang=en. The error description of the
LIDAR  data  can  be  found  at  https://www.maanmittauslaitos.fi/en/maps-and-spatial-
data/expert-users/product-descriptions/elevation-model-2-m. The quasigeoid heights was
downloaded  from  https://www.maanmittauslaitos.fi/kartat-ja-paikkatieto/asiantuntevalle-
kayttajalle/koordinaattimuunnokset. The MODIS/Terra Snow Cover Daily L3 Global 500 m
SIN Grid, Version 6 data is available at https://nsidc.org/data/MOD10A1/versions/6. The
OpenStreetMap road network can be acquired at https://overpass-turbo.eu/. The building
density  information  of  the  city  of  Helsinki  can  be  found  at
https://hri.fi/data/en_GB/dataset/rakennustietoruudukko. The LISFLOOD-FP  model  is
available        for        non-commercial        research        purposes        from
https://zenodo.org/record/4073011#.YeWAdP7P2Ul. The Bare-earth ArcticDEM can be
accessed at https://doi.org/10.5523/bris.3c1l2q7u1x14a262m6z7hh0c4r. The PMF algorithm
can        be        accessed        at
http://www.pylidar.org/en/latest/_modules/pylidar/toolbox/grdfilters/pmf.html, the  SMRF
algorithm can be accessed at https://github.com/thomaspingel/smrf-matlab.

**Author contributions**

Yinxue Liu wrote the manuscript and carried out the data processing and analysis. Paul Bates
and Jeffery Neal provided comments on various drafts as well as advised on the analysis work.

**Competing interests**

The authors declare that there is no conflict of interest.

**Acknowledgements**

Yinxue Liu was supported by the China-Scholarship-Council (CSC) – University of Bristol
Joint PhD Scholarships Program. Paul Bates was supported by a Royal Society Wolfson
Research Merit award and UK Natural Environment Research Council grant NE/V017756/1.
Jeffrey Neal was supported by NE/S006079/1.

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
