# Peer review of "Bare-earth DEM Generation from ArcticDEM, and Its Use in Flood Simulation"

_Natural Hazards and Earth System Sciences, 2022_

## Author Comment (AC1)

**RC1**: 'Comment on nhess-2022-210', Dai Yamazaki, 21 Sep 2022

\<General Comments\>

The proposed research compared the different surface object removal method on Arctic DEM, and analyzed how DEM error correction impacts flood inundation simulation. Considering increasing availability of high-resolution DEM, I think the suggestions from this research (which correction method is better, which parameters are feasible, how correction impact flood inundation simulation) is very useful. It is good to see that the DEM error correction using SMRF method is robust using wide range of parameters.

One major suggestion I'd like to provide to enhance the manuscript is to include more discussion on the transferability of the proposed method (SMRF algorithm with optimum parameter range) to other regions. Readers must be interested in whether the optimum SMRF parameters detected by this study can be safely used to other regions or not. Please include some discussions about the parameter transferability (detailed suggestions are in the Specific Comments.)

Other than the above concern, the manuscript is I think very well organized. And it can be accepted after minor revision.

Thank you so much for your thorough comments, Dr Yamazaki. They are very useful in improving our manuscript. Our responses to each comment are made below.

\<Specific Comments\>

Line 41: "exponentially increased computational costs".

"Exponentially" is not precise. The calculation cost of 2D flow simulation follows approximately $(1/dx)^3$ where dx is the special resolution. If grid size becomes half, the computational cost is almost 8 ($=2^3$) times, it's not exponential.

Thanks for pointing this out. We revised the writing to be more precise.

'*because of the rapidly increased computational cost of running fine grid models (roughly the ratio of grid change to the power of three)*'.

Line 78: ASTER GDEM

Why not including AW3D DEM in reference here, which is more preceise and now being widely used as high-accuracy stereo-view DEM?

This is a good point. We replaced the ASTER with the AW3D in the revision.

L84: SETSM

Please explain what "SETSM" stands for? When it appears first time.
Thanks for pointing this out. We changed it in the revision.

'*ArcticDEM covers areas above 60°N and was produced using the Surface Extraction with TIN-based Search-space Minimization (SETSM) method from in-track and cross-track high-*

*resolution (~0.5 m) imagery acquired by the WorldView and GeoEye satellites.'*

P257: Table 1

I suggest to put a line to distinguish PMF and SMRF, as the boundary is not clear.

Good suggestion. We now revised the table as below.

*Table 1.* *Key parameter settings of the morphological filters tested in the three samples.*

| Filter | Sample | Key Parameters | | | |
| | | Window size (m) | | Slope threshold | |
| | | range | interval | range | interval |
| --- | --- | --- | --- | --- | --- |
| PMF | S1 | 10-66 | 4 | 0.1-0.3 | 0.2 |
| | S2 | 10-66 | 4 | 0.1-0.3 | - |
| SMRF | S1 | 10-50 | 2 | 0.01-0.1 | 0.005 |
| | S2 | 10-50 | 2 | 0.01-0.1 | 0.005 |
| | S3 | 10-180 | 10 | 0.03-0.15 | 0.01 |

* The unit of the slope threshold values shown here is radian for PMF, percent of slope/100 for SMRF.

P270: Replaced with the LIDAR DTM values.

Please describe the situation of the Arctic DEM original value here. Are they "missing data", or there are large error?

Thanks for this comment. This was done purely because that the errors in these areas are not of concern of this paper as they are covered by water, whereas we focus on modelling the flooding over the land surface in this paper.

P278: ensuring that the difference between the simulations was distinguishable.

The logic here is unclear. If large-magnitude flood is used as a test case, I assume flood extent is more confined by large-scale topography. Focusing on smaller-magnitude flood might be better to discuss the impact of topography improvement on flood risk estimation.

Thanks for raising this point. Regarding to flood performance evaluation, the flood inundation extent and water depth are the two aspects to our concern.

From the perspective of flood extent, we agree that larger-magnitude floods are more confined by large-scale topography compared to small ones. We also agree that the inundation extent of smaller-magnitude flood will be more sensitive to the topography error. But the flood extent can become overly sensitive to the topography error when the inundation depth is extremely shallow.

From the perspective of the water depth error, larger magnitude of scenarios can better assess the impact of topography error on the simulated water depth as the inundation will cover a higher ratio of the overall areas.

We meant to emphasize that this rarely occurred frequency of 500 years extreme rainfall was used as we preferred a short duration scenario. The sentence is now revised as '*A designed rainfall scenario with duration of 3 h and return period of 500 years was used in the simulation. To minimize the simulation time a short duration scenario is preferred, which led to our choice of the 3 h duration. The relatively low occurring frequency (500 years return period) was then chosen to avoid flood inundation being overly sensitive to the topography which would happen when the inundation is extremely shallow.*'

We also want to point out that the return period of 500 years is for the designed rainfall. It indicates the amount of water comes in the simulated domain but does not necessarily indicate the same return period of flooding.

P286: small isolated wet areas

Please explain the mechanism of how these are caused?

Thanks for the comment. The small isolated wet areas are common in pluvial flooding because its flow pathways are often not so confined as fluvial flooding. These small, isolated areas also appear in the simulation results of the LIDAR data. We excluded these areas with the same method for the LIDAR and ArcticDEM simulations.

P347: Figure 4

I don't think the cross marks for man error (right column) are meaningful. The optimum points exist as a "line" in white-color area, rather than as a point in case of the mean error. Putting one cross mark could be miss-leading.

Thanks for this comment. Yes, there might be multiple points where the absolute value of the mean error is the smallest among the 234 realizations. To avoid miss-leading, we now change the mark for the mean error as the location where the lowest RMSE is achieved (shown below). The figure caption is changed correspondingly.

[Figure]

**Figure 4**. Surface plots of the slope threshold and the window size parameters of the SMRF filter against the RMSE, the RMSE reduction percentage and Mean error of the filtered DEM-ArcticDEM-SMRF for sample S3. The location of the smallest values of the RMSE (which is the same as the location of the greatest values of the RMSE reduction) are marked as ×, with the values displayed. The values of the Mean error at the above location are displayed and marked as +. Parameter details can be found in Table 1.

L361: More than 40% of the parameter combinations can 362 reduce the RMSE by greater than a half.

This is important, but there must be something more to discuss for ensuring the robustness of the method. [1] The optimum combinations are almost same for three different land covers, suggesting the robustness of the parameter for various-time land-surface characteristics. [2] The skill-score does not significantly drop when parameter combination is slightly changed from the optimum location, suggesting the robustness of

estimated parameters. There must suggest the transferability of the method to another region?

Thanks for offering this comment. We made changes to the results section and extend the robustness in the discussion section in the revision of the manuscript.

*'These optimum parameters only vary within a rather small range for different land covers, suggesting that the parameter choice is robust for various land-surface characteristics. Moreover, the error removal effectiveness does not significantly drop when parameters slightly deviate from the optimum location, suggesting the robustness of parameters. More than 40% of the 234 parameter combinations can reduce the RMSE by greater than a half. The robustness of the filter across different land covers and a range of parameters is favoured as this loosens the prior knowledge request of the study site and simplifies the parameter setting for application across large domains.'*

This robustness is inherent in the SMRF algorithm and thus is transferable. The robustness of parameter to the error reduction was also demonstrated by the Pingel (2013) which proposed the SMRF filter. The key of ensuring the robustness is deciding the range of the parameters. We explained it below.

In theory, to remove all objects in the target areas the window size should correspond to the size of the largest object in the target area. This is a hypothesized entirely flat area. In a real topography over large areas, there is always hilly areas or terrain variations. Therefore, applying such a window size will identify some hilly areas as objects incorrectly and flatten them, which will result in negative errors in the filtered DEM. Therefore, a smaller window size has to be chosen instead. This smaller window size will inevitably miss out some of the object with large sizes. Similarly, the choice of the slope threshold has to consider preserving the hilly areas (using large slope threshold) and removing the objects (using small slope threshold).

The surface plot of the RMSE and Mean error of the filtered ArcticDEMs (Figure 4) evidence that the combination of large window size and small slope threshold enforces the strictest rule of removing objects while the combination of large window size and large slope threshold enforces the loosest rule. Therefore, we are confident that the robustness and the pattern of the filtered DEM error and the parameter choice is transferable to other sites (although the parameter numbers might be different in other sites, and we will explain finding the optimum parameters in the response for comment L589).

L379: The error distribution of…

Please connect this sentence to the following paragraph. One sentence paragraph is not recommended.

Thanks for this comment. We changed this in the revision.

L382: Figure 5

The blue color overlaid on satellite map is very difficult to see. Please adjust colors.

Thanks for this comment. We have changed Figure 5 as below.

[Figure]

L405: Figure 6:

Can you add one more "cross symbol" which represent the result of the best optimum parameter combination (which are common in all land covers). Readers must be interested in how "best-corrected DEM performs" simultaneously for all skill scores for all land covers.

Thanks for this comment. We add the mark for the lowest RMSE as triangle now. We now also mark the location where the ArcticDEM-SMRF were filtered with the window size of 10 m, which corresponds to the comment to L450.

[Figure]

**Figure 6**. Surface plot of the CSI score, Hit Rate, FAR, the water depth RMSE and Mean error (ME) simulated using the ArcticDEM-SMRF realizations (ArcticDEM filtered using the 234 SMRF parameter combinations) at sample S3 plotted against the RMSE and the Mean error of each realization member. The location of the highest CSI and Hit Rate, the smallest FAR, RMSE and the smallest absolute value of mean water depth error are marked as red crosses, with the values displayed. The location of the lowest RMSE of the ArcticDEM-SMRF

are marked as triangle, with values displayed (values are not shown if both the location and value are the same as the best flood inundation metric value). In addition, the RMSE, Mean error of the original ArcticDEM are located and marked as blue crosses in each panel with the five metrics value of the original ArcticDEM simulation displayed. Locations of ArcticDEM-SMRF filtered with window size = 10 m are marked with cyan color.

L446: ArcticDEM-SMRF with larger error.

What do you mean by "error" is not clear here. Do you mean "larger elevation error"?

Thanks for this comment. The error here meant to say that the filtered ArcticDEM with greater error (RMSE) than the optimum ArcticDEM-SMRF. The optimum ArcticDEM-SMRF is defined as the ArcticDEM-SMRF with the smallest RMSE among the 234 ArcticDEM-SMRF realizations. We now make this clearer as '*However, there are two other cases where equally good flooding performance can be simulated using ArcticDEM-SMRF with larger error than the optimum ArcticDEM-SMRF.*'

L450: shown at the spike areas in Fig6

I cannot find where is "the spike" in Figure 6. Please provide better explanation.

Thanks for this comment. We revise the Figure 6 by adding marks to the area (see the revised Figure 6 in the response to L405). This area means that under-filtered ArcticDEM with good scores of assessed metrics.

L458: Figure 7

Please check the color map of the bottom figure. It seems there are many "green" colors which is not in the color bar.

Thanks for this comment. The previous figure display was log normalized which was not correctly reflected in the colour bar. We now revise Figure 7 as below. It should be noted that this display change does not alter our analysis of water depth error in the manuscript.

[Figure]

L517: similarly good flood simulation

I agree that the flood extents are almost similar, but how about flood depth? Can we say "similar"?

This is a good point. The water depth errors of the over-filtered ArcticDEM-SMRF and the under-filtered ArcticDEM-SMRF also compare well with the simulations based on the ArcticDEM-SMRF with the lowest RMSE error as shown in the Figure 6 row 4 and row 5.

We further checked the error of the water surface elevation (WSE) of these simulations and found that the over-filtered ArcticDEM-SMRF WSE error remain as similar low as the simulation of the ArcticDEM-SMRF with the lowest RMSE error, but not the under-filtered ArcticDEM-SMRF. We now add this finding of the water surface elevation error in the discussion section 5.2.

L543: ICESAT2

It should be "ICESat-2".

Thanks for this comment. The word will be changed to ICESat-2.

L589: which resulted in an optimal window size of 30 m and slope threshold of 0.07 in the city of Helsinki.

Please make some discussions on the possibility of transferring this parameter to other regions, or possibility of estimating best parameter for other regions (without Lidar DEM coverage). Readers must be interested in this. If the parameter has relationship to land object characteristic (such as typical building size), there might be a chance to find good parameters for other regions.

As we analyzed above (response to comment L361), the residual errors mainly depend on the ratio of areas with residual errors and error values of these areas. The areas of residual errors mostly are contributed by objects that the used parameter cannot remove and by hills that were flatten incorrectly with the used parameters. Thus, balanced window size and slope threshold between the minimum and maximum as the SMRF parameters are needed to achieve the bare-earth DEM with the lowest error.

In this paper, we found a range of 0.04-0.1 of the slope thresholds has overall good performance of filtering the ArcticDEM, with 0.07 (or 7%) generating the bare-earth ArcticDEM with the lowest error. The value 0.07 is close to the mean slope in our study site (0.077 or 7.7%). Note the slope is calculated as percentage.

The 30 m window size are gauged visually corresponding to the average size of the objects because of lacking building footprint data. We tried to compute the object sizes from the difference between the original ArcticDEM and the LIDAR DTM. But we found the footprint heavily depend on the elevation difference threshold and the smallest patch size threshold. This makes the size measurements derived from that being not very reliable. So, we did not include the size analysis from this approach in the manuscript.

Therefore, we suggest testing the mean value of the slope and average size of objects first and adjusting these values up and down will likely find the optimum parameter quickly. Discussion of the parameter choice is now extended in the revision of Section 5.2.

The theory of SMRF algorithm and the optimum window size between different land covers clearly show that there is a positive relationship between the optimum window size and the size of the objects. But because of lacking the footprint data of the objects we could not further quantify this relationship. The slope of study site will also play a role in this relationship. Thus, it might not be easy to define a universal relationship with this study site. We argue that further quantification might not be so critical. Because that both urban and forest sizes are not very difficult to gauge given general knowledge, and that the error is robust to a reasonable range of the parameters.

---

## Author Comment (AC2)

**RC2**: 'Comment on nhess-2022-210', Guy J.-P. Schumann, 06 Nov 2022
This paper is a comparison of the ArcticDEM vs LiDAR for urban flood simulation which uses Helsinki as an example case.

The paper is generally well written and follows a clear structure. The methodology used is sound and fairly straightforward. The results are well presented.

This type of analysis is quite timely as there are at present substantial efforts and initiatives under way to get better accuracy global DEM data sets and a DEM like the ArcticDEM may become available sson for global low-lying lands.

Thank you so much Dr Schumann for reviewing our paper, the kind words, and the helpful comments.

In my opinion this paper can be accepted for publication after some minor points are addressed:

- Please verify that referring to DigitalGlobe is correct or should it be Maxar?

We are aware of that the DigitalGlobe was acquired by Maxar in 2017, but in this paper, we followed the term used in the Polar Geospatial Center (where ArcticDEM was distributed) as DigitalGlobe.

- It seems to me that the vertical error of the bare earth ArcticDEM in the urban area is about 0.5 m and the simulated water depth RMSE is almost double. If this is correct, could the authors comment on this in the context of whether this type of water depth RMSE in urban areas is still acceptable?

Thank you for this comment. Actually, it is the other way around. As shown in Figure 6, the generated bare-earth ArcticDEM has a RMSE error of 1.02 m (the lowest one). Using this bare-earth ArcticDEM the simulated water depth error (RMSE) is 0.3 m. This can be caused by that the error values of the ArcticDEM-SMRF in inundated areas or possibly inundated areas are likely smaller than the numbers reported for the overall areas. Because most of the residual errors are in hilly areas that were flattened incorrectly and forest areas with large patch sizes, while these areas have relatively small overlap with the inundated areas.

- It would be useful I think if the authors could comment on the resolvability of individual buildings within the ArcticDEM - I imagine some kind of density measure should allow a comparison between LiDAR DSM and ArcticDEM DSM, the results of which could explain the significant differences in water depth RMSE obtained. Maybe some kind of DSM surface roughness measure comparison.

We appreciate this comment. In this manuscript, we used the LiDAR DTM as the reference DEM input instead of the DSM. We used DTM because that we chose to build our model at 10 m (which is likely greater than typical building gaps) considering the computational cost of simulating all the 234 ArcticDEM realizations in an area of 192 km$^2$.

Analyzing the resolvability of ArcticDEM of individual buildings would be very interesting. However, the building footprint data is not publicly available for the city of Helsinki.

Neither does the DSM of the same spatial resolution of the LIDAR DTM (2 m). If the building footprint or the DSM at 2 m or better spatial resolution becomes available, this will allow analyzing the flood inundation performance by linking to the building resolvability and surface roughness.

We tried to compute the building footprint from the difference between the original ArcticDEM and the LIDAR DTM. But we found the derived patch sizes heavily depend on the elevation difference threshold and the smallest object size threshold.

We showed an example here using the difference with the elevation difference threshold as 1 m (i.e., the ArcticDEM-original-LIDAR >1 m as objects) and the smallest object size as 1 pixel (Figure below). We found that the narrow streets between the buildings are not captured in the ArcticDEM, which is obvious for sample S2 (demonstrated in the Figure below). The filter can identify these as objects and flattens the streets to the same level as the adjacent buildings.

Figure. The patches (pink in 50% transparency) defined by the ArcticDEM-LIDAR>1m overlay the original ArcticDEM of sample S1 and S2.

**ArcticDEM-original-LIDAR>1m**
**(50% transparency)**

[Figure]

[Figure]

- Could the authors comment on how transferable their presented method and error statistics would be to other urban use cases.

The robustness and the pattern of parameter response to the error of the filtered DEM is transferable to other study sites as this is inherent in the filter algorithm. From both the theory of the algorithm and our result, we argue that starting from the mean values of the artefact sizes and slope threshold and varying them up and down has a greater chance to find the optimum parameter combinations quickly. Please refer to details in our response to the comment L361, L589 above.

The error statistics will likely change depending on the study site. There are two scenarios that the error might be larger than the values reported in this study. First, if the study site has a high ratio of artefacts with large sizes (much larger than typical building sizes such as closed forest canopy), both the identification of these objects and the interpolation of the terrain in these areas will likely introduce some errors. Second, if the study site has a high ratio of hilly areas, distinguishing them from objects identification

will be difficult and the residual error of the filtered DEM will likely be greater than reported in this paper. We include this in the discussion and conclusions.